

# Association between the degree of fibrosis in fibrotic focus and the unfavorable clinicopathological prognostic features of breast cancer

Yongfu Li[1,2], Yuhan Wei[1], Wenjun Tang[2], Jingru Luo[2], Minghua Wang[3], Haifeng Lin[2], Hong Guo[4], Yuling Ma[5], Jun Zhang[6] and Qin Li[1]

[1] Department of Oncology, Beijing Friendship Hospital, Capital Medical University, Beijing, China
[2] Department of Oncology, The Second Affiliated Hospital of Hainan Medical University, Haikou, Hainan Province, China
[3] Department of Pathology, The Second Affiliated Hospital of Hainan Medical University, Haikou, Hainan Province, China
[4] Department of Surgery, Beijing Changping District Hospital of Traditional Medicine, Beijing, China
[5] Department of Gynecology, Beijing Changping District Hospital of Traditional Medicine, Beijing, China
[6] Department of Hematopathology, University of Texas MD Anderson Cancer Center, Houston, TX, United States of America

Corresponding author
Qin Li, oncologistinbj@163.com

## ABSTRACT

**Objective**. To explore the association between the degree of fibrosis in fibrotic focus (FF) and the unfavorable clinicopathological prognostic features of breast cancer.

**Methods**. A total of 169 cases of breast invasive ductal carcinoma (IDC) were included in the study. Hematoxylin and eosin (H&E) staining was performed in the primary lesion of breast IDC and the degree of fibrosis in tumor-stromal FF was assessed. The association between the degree of fibrosis in FF and the well-known clinicopathologic features of breast cancer was investigated and the influence of the degree of fibrosis in FF on the survival was analyzed.

**Results**. Tumor size >2 cm ($P = 0.023$), vascular invasion ($P = 0.011$), lymphatic vessel invasion ($P < 0.001$) and HER-2+ ($P = 0.032$) were positively correlated with the degree of fibrosis in FF in breast IDC. The result of multivariate analysis showed that lymphatic vessel invasion was the only independent correlation factor of high fibrosis in FF in breast IDC (OR = 3.82, 95% CI[1.13 ∼ 12.82], $P = 0.031$). The Nottingham prognostic index (NPI) of high fibrosis in FF was significantly higher than that of mild and moderate fibrosis in FF in the no vascular infiltration subgroup, the no nerve infiltration subgroup, and the Luminal A subgroup (P = 0.014, 0.039, and 0.018; respectively).

**Conclusions**. The high fibrosis in FF is closely associated with the strong invasiveness and the high malignancy of breast IDC. The degree of fibrosis in FF might be considered as a very practical and meaningful pathological feature of breast cancer.

## INTRODUCTION

Breast cancer is the most common malignant tumor in women across the world (*DeSantis et al., 2017*). The prognosis of breast cancer is closely associated with various clinicopathological characteristics, such as age, tumor size, pathological type, lymph node metastasis status, histological grade, lymphovascular invasion, Ki-67 index, hormone receptor (HR) status, human epidermal growth factor receptor-2 (HER-2) expression, etc. These well-known pathological features have been widely used to make clinical treatment plans and predict the prognosis of breast cancer. However, the pathophysiology of breast cancer is not only closely related to the tumor cells, but also related to the tumor microenvironment which is composed of sromal cells, infiltrating immune cells, vasculature, extracellular matrix, and various cell signaling factors. The tumor microenvironment plays an important role in the genesis, development, invasion, metastasis, immune escape and chemotherapy resistance of tumor (*Reisfeld, 2013*). FF is a pathological change in the tumor microenvironment of breast cancer, with an incidence of 18.7–53.0% (*Colpaert et al., 2001*; *Hasebe et al., 1996*; *Kornegoor et al., 2012*; *Mujtaba et al., 2013*; *Van den Eynden et al., 2008*). *Hasebe et al. (1996)* described the pathologic features of FF in the breast cancer stroma in detail for the first time: a fibrotic lesion, with an appearance of scar or radiating expanding fibrosclerotic core, almost located in the center of carcinoma and consisted of variable amounts of collagen fibers and fibroblasts. At present, it is believed that the formation of FF is driven mostly by intratumoral hypoxia, which reflects the malignancy of carcinoma. Thus, FF is considered as a very practical and easily assessable clinicopathological parameter in breast cancer (*Baak et al., 2005*; *Hasebe et al., 2000*; *Hasebe et al., 2002*; *Hasebe et al., 1997*; *Maiorano et al., 2010*; *Van den Eynden et al., 2007*). According to the variable proportions of collagen fibers and fibroblasts, the degree of fibrosis in FF is classified into three categories: mild, moderate, and high (*Hasebe et al., 1996*; *Van den Eynden et al., 2007*). The fruit of much research has been presented in the correlation between FF and the poor outcome in breast cancer. However, little is known about the association between the degree of fibrosis in FF and the prognosis of breast cancer so far. Whether the degree of fibrosis in FF could better reflect the prognosis of breast cancer than FF has become a research hotspot. Therefore, a retrospective study was conducted to explore the association between the degree of fibrosis in FF and the unfavorable clinicopathological prognostic features of breast cancer.

## MATERIAL AND METHODS

### Cases

The objects of this study were 169 patients with primary breast cancer who underwent surgeries between January 1, 2016 and December 31, 2018 at the Second Affiliated Hospital of Hainan Medical University. Inclusion criteria: (1) female diagnosed with primary breast invasive ductal carcinoma (IDC); (2) IDC must be the principal component (>50%) in mixed pathological type; (3) no distant metastasis; (4) no neoadjuvant therapy before surgery; (5) FF stained with H&E was present in the primary lesion of breast cancer. Exclusion criteria: (1) male breast cancer; (2) infiltrative specific breast cancer; (3)

secondary carcinoma of mammary gland. All included cases were followed up once every three months by telephone.

## Ethical

The Second Affiliated Hospital of Hainan Medical University granted Ethical approval to carry out the study within its facilities (Application Ref: LW005). Our Institutional Review board waived the written consent, and the verbal consent was used.

## Materials

Personal information of included cases was collected accurately, including age, family tumor history, smoking history, and drinking history. The clinicopathological features were evaluated based on the pathological reports and the medical records, including tumor size, regional lymph node metastasis status, stage, histological grade, vascular invasion, lymphatic vessel invasion, nervous infiltration, estrogen receptor (ER) status, progesterone receptor (PR) status, ki-67 index, HER-2 expression, etc. The measurement of tumor size was based on the largest diameter of invasive component in histological sections. The largest diameter of metastatic lesion >0.2 mm in axillary lymph node was defined as regional lymph node metastasis (*Plichta et al., 2018*). Disease stage was assessed according to the eighth edition of American Joint Committee on Cancer Staging Manual for breast cancer (*Plichta et al., 2018*). Histological grade was evaluated according to Nottingham modification of the Scarff-Bloom-Richardson histological grading system for invasive breast cancer (*Elston & Ellis, 1993*). Stained with immunohistochemistry (IHC), the ER expression rate of tumor cells $\geq$ 1% was defined as positive ER expression (ER+), otherwise it was defined as negative ER expression (ER-). This criterion also was applied to the evaluation of the PR expression (PR+/-). The positive PR expression was classified into two categories: low (<20%) and high ($\geq$ 20%) (*Prat et al., 2013*). Positive hormone receptor expression (HR+) was defined by ER+ or/and PR+. In the selected area of slides with >500 tumor cells, the percentage of tumor cells with positive Ki-67 expression (IHC) was defined as Ki-67 index. Ki-67 index was also classified into two categories: low ($\leq$ 14%) and high (>14%) (*Goldhirsch et al., 2011*). HER-2 expression status was defined according to the recommendations for HER-2 testing in breast cancer and classified into two categories: positive (HER-2+) and negative (HER-2-) (*Wolff et al., 2013*). Combining the immunohistochemical findings of ER, PR, HER-2, and Ki-67 index, breast cancer was classified into four molecular subtypes including Luminal A, Luminal B, HER-2 overexpression, and triple negative breast cancer (TNBC). NPI is a widely accepted clinicopathological scoring system for early breast cancer prognostication. NPI = Tumor size (cm) ×0.2 + Histological grade (1~3) + Lymph node stage (1~3) (*Lee & Ellis, 2008*). Histological grade 1~3 was scored as 1~3 points, respectively. Lymph node stage 1 meant no node involved. Lymph node stage 2 meant 1~3 low axillary nodes involved or internal mammary node involved. Lymph node stage 3 meant $\geq$ 4 low axillary nodes involved and/or the apical axillary nodes involved or both low axillary nodes and internal mammary nodes involved. Lymph node stage 1~3 was scored as 1~3 points, respectively.

## Methods

All specimens of the primary lesion of breast cancer were fixed in 10% formalin and cut into 5 $\mu$m-thick sections which were stained with H&E and analyzed by microscopic examination. Firstly, the location, size, appearance and components of FF should be observed under low power microscope. FF was mostly located in the center of the tumor. The size of FF was $\geq$ one mm, otherwise it could not be defined as FF. FF usually appeared as scar-like lesion (Fig. 1A) or irregular moth-eaten radiating fibrosclerotic core (Fig. 1B). Numerous tumor cells could be observed frequently around FF. Besides, linear-growing tumor cells and tumor nests could be usually found in FF with diameter >three mm. Coagulative necrosis could be present in some FFs. But coagulative necrosis without collagen fiber deposition and fibroblast proliferation was insufficient to be called FF. Sometimes, haemorrhage could be found in FF. Secondly, the components of FF were observed under high power microscope. FF was mainly composed of different proportions of fibroblasts and collagen fibers. The collagen fibers stained with H&E were thick and arranged closely in bundle. Different amounts of fibroblasts proliferating abnormally could be found among the collagen fibers. In addition, the densities of micro-vessels and micro-lymphaticvessels in FF were significantly higher than those in normal tissue. Evaluating the degree of fibrosis in FF is the last but most important step. According to the proportion of fibroblasts and collagen fibers, FF was classified into three semi-quantitative categories (*Van den Eynden et al., 2007*): (1) mild fibrosis meant that FF consisted of a large number of fibroblasts and small amount of collagen fibers; (2) moderate fibrosis intermediated mild fibrosis and high fibrosis; (3) high fibrosis meant that FF was mainly composed of collagen fibers. Two experienced pathologists were involved in pathological examination and the other senior pathologist should reassess the degree of fibrosis in FF when the two pathologists did not agree on the conclusion.

## Statistical analysis

Statistical analyses were performed using IBM SPSS statistics software version 25.0. The mean age and the mean NPI were described by $\bar{X} \pm$SD. Qualitative data of general and clinicopathologic features were described by case number, rate, and constituent ratio. Lower quartile ($P_{25}$), median (M), and upper quartile ($P_{75}$) were used to describe the distribution of NPI variables in different degrees of fibrosis in FF. The associations between the degree of fibrosis in FF and clinicopathologic features were analyzed by Mann–whitney $U$ test and Jonckheere-Terpstra test. After stratificating the patients by clinicopathologic features, most NPI variables were skewed distribution. The Mann-whitney U test was performed to analyze the difference of NPI variables between mild and moderate fibrosis in FF and high fibrosis in FF. The factors significantly associated with the degree of fibrosis in FF in the univariate analyses ($P < 0.05$) were entered together into ordinal logistic regression analysis. The remaining factors in the multivariate analysis were significant at $P < 0.05$. Generalized linear model was used to calculate the odds ratio (OR) and 95% confidence interval (95% CI). All analyses were two-sided and $P < 0.05$ was considered to indicate a statistically significant difference.

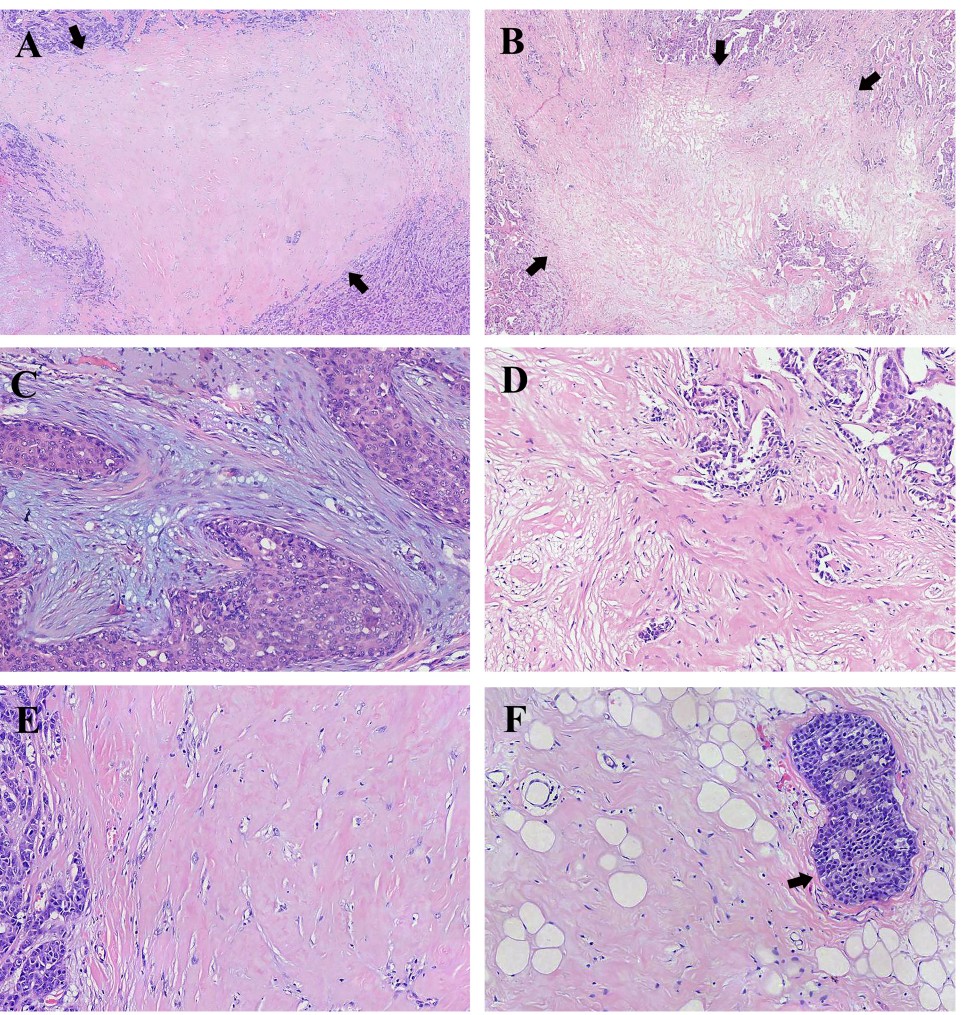

**Figure 1  Representative histology of fibrotic focus (FF) in breast invasive ductal carcinoma (IDC).** (A) A representative histology of FF with the appearance of scar-like lesion indicated by arrows (H&E; magnification, ×20). (B) A representative histology of FF with the appearance of irregular moth-eaten radiating fibrosclerotic core indicated by arrows (H&E; magnification, ×20). (C) FF with mild fibrosis showing high number of fibroblasts and small amount of collagen fibers in stroma (H&E; magnification, ×100). (D) FF with moderate fibrosis intermediating between mild fibrosis and high fibrosis and numerous tumor nests in FF (H&E; magnification, ×100). (E) FF with high fibrosis showing mostly hyalinized collagen fibers (H&E; magnification, ×100). (F) High fibrosis and peripheral vascular invasion indicated by the arrow (H&E; magnification, ×100). Photograph credit: Doctor Xiangtao Lin.

## RESULTS

### General characteristics of all cases

Of the 169 cases, the mean age was $51.6 \pm 10.0$ (range, 28.0–80.0). 6 cases (3.6%) had family tumor histories, 21 cases (12.4%) had drinking histories, and none had smoking history.

**Table 1   The characteristics of patients.**

| Clinicopathologic variables | n (%) | Clinicopathologic variables | n (%) |
|---|---|---|---|
| **Age (years)** | | **Nervous infiltration** | |
| ≤ 50 | 79 (46.7) | Yes | 32 (18.9) |
| >50 | 90 (53.3) | No | 81 (47.9) |
| **Tumor size** | | Unknown | 56 (33.2) |
| ≤ 2 cm | 59 (34.9) | **ER** | |
| >2 cm | 109 (64.5) | Positive | 108 (63.9) |
| Unknown | 1 (0.6) | Negative | 60 (35.5) |
| **Lymph node** | | Unknown | 1 (0.6) |
| **metastasis** | | **PR** | |
| Yes | 90 (53.3) | Positive | 100 (59.1) |
| No | 76 (45.0) | Negative | 68 (40.2) |
| Unknown | 3 (1.7) | Unknown | 1 (0.7) |
| **stage** | | **Ki-67** | |
| I | 34 (20.1) | ≤14% | 35 (20.7) |
| II | 87 (51.5) | >14% | 129 (76.3) |
| III | 44 (26.0) | Unknown | 5 (3.0) |
| Unknown | 4 (2.4) | **Her-2** | |
| **Histological grade** | | Positive | 36 (21.3) |
| I ∼ II | 88 (52.1) | Negative | 95 (56.2) |
| III | 62 (36.7) | Unknown | 38 (22.5) |
| Unknown | 19 (11.2) | **Molecular subtype** | |
| **Vascular invasion** | | Luminal A | 16 (9.5) |
| Yes | 67 (39.6) | Luminal B | 93 (55.0) |
| No | 60 (35.5) | HER-2 | |
| Unknown | 42 (24.9) | overexpression | 17 (10.1) |
| **Lymphatic vessel** | | TNBC | 24 (14.2) |
| **invasion** | | Unknown | 19 (11.2) |
| Yes | 51 (30.1) | | |
| No | 64 (37.9) | | |
| Unknown | 54 (32.0) | | |

## Clinicopathological characteristics of all cases

The main pathological pattern was IDC. Thirteen cases (7.7%) were mixed with one kind of infiltrative specific breast cancers, including mucinous carcinoma (three cases, 1.8%), lobular carcinoma (two cases, 0.12%), invasive papillary carcinoma (two cases, 0.12%), apocrine carcinoma (two cases, 0.12%), medullary carcinoma (one case, 0.6%), neuroendocrine carcinoma (one case, 0.6%), pleomorphic carcinoma (one case, 0.6%), and basal carcinoid carcinoma (one case, 0.6%). Other clinicopathologic characteristics are presented in Table 1.

## The degree of fibrosis in FF

FF had been observed in the primary lesion of each case. Eleven cases (6.5%), 65 cases (38.5%), and 93 cases (55.0%) were evaluated as mild, moderate, and high fibrosis, respectively. Representative histology of FF in breast IDC is shown in Figs. 1C–1E.

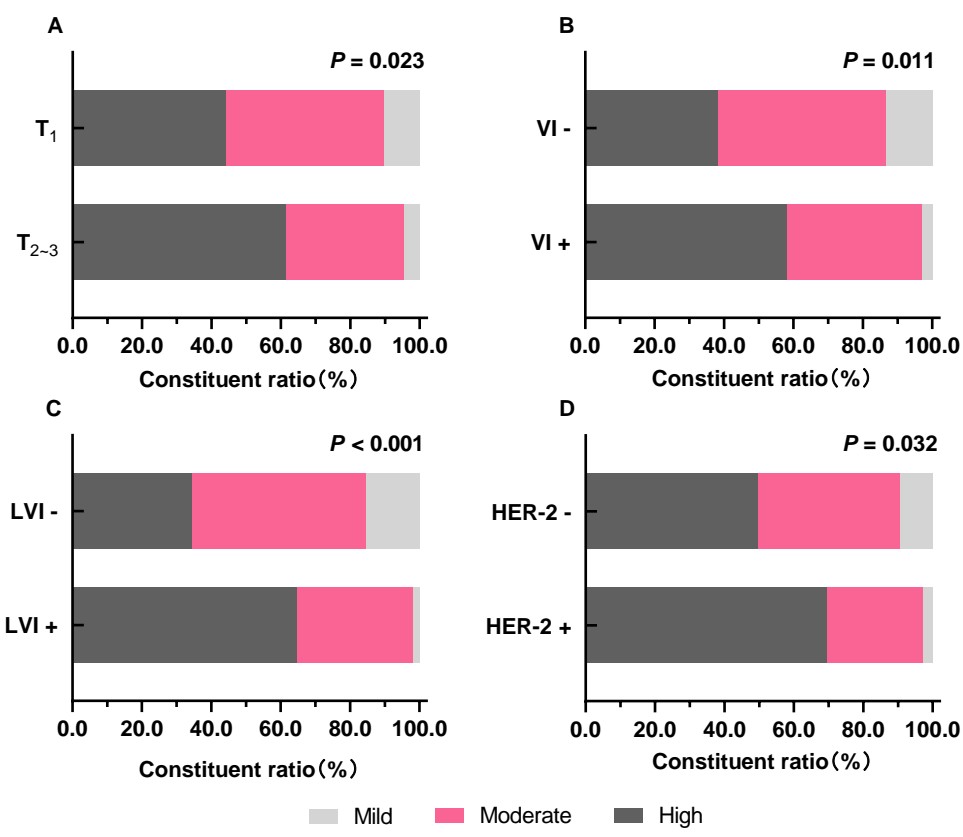

**Figure 2 The distribution of different degrees of fibrosis in FF in different clinicopathologic features.** (A) The comparison of the proportions of different degrees of fibrosis in FF between T ≤ two cm (T₁) and T > two cm (T₂₃). (B) The comparison of the proportions of different degrees of fibrosis in FF between VI- and VI+. (C) The comparison of the proportions of different degrees of fibrosis in FF between LVI- and LVI+. (D) The comparison of the proportions of different degrees of fibrosis in FF between HER-2- and HER-2+. Notes: T, tumor size; VI-, no vascular invasion; VI+, vascular invasion; LVI-, no lymphatic vessel invasion; LVI+, lymphatic vessel invasion.

## Correlation between the clinicopathological characteristics and the degree of fibrosis in FF

Tumor size >two cm, vascular invasion, and lymphatic vessel invasion were significantly correlated with high fibrosis in FF (Figs. 1F and 2A–2C). Age, lymph node metastasis, stage, histological grade, and nerve infiltration were not significantly associated with the degree of fibrosis in FF. Correlation between the clinicopathological characteristics and the degree of fibrosis in FF is presented in the Table 2.

## Association between molecular subtypes and degree of fibrosis in FF

Significant association was observed between HER-2+ and high fibrosis in FF (*P* = 0.032) and shown in Fig. 2D and Table 3. The other sub-factors of molecular subtypes (ER, PR, and Ki-67 index) and molecular subtypes were not significantly associated with the degree of fibrosis in FF (Table 3).

**Table 2  The correlation between clinicopathological characteristics and degree of fibros.**

| Clinicopathologic variables | Degree of fibrosis in FF | | | Z/J-value | P-value |
|---|---|---|---|---|---|
| | Mild n (%) | Moderate n (%) | High n (%) | | |
| **Age** (n = 169) | | | | | |
| ≤50 years | 5 (45.5%) | 32 (49.2%) | 42 (45.2%) | −0.397[a] | 0.691 |
| >50 years | 6 (54.5%) | 33 (50.8%) | 51 (54.8%) | | |
| **Tumor size** (n = 168) | | | | | |
| ≤2 cm | 6 (54.6%) | 27 (42.2%) | 26 (28.0%) | −2.276[a] | 0.023 |
| >2 cm | 5 (45.4%) | 37 (57.8%) | 67 (72.0%) | | |
| **Lymph node metastasis** (n = 166) | | | | | |
| No | 6 (54.5%) | 34 (53.1%) | 36 (39.6%) | −1.746[a] | 0.081 |
| Yes | 5 (45.5%) | 30 (46.9%) | 55 (60.4%) | | |
| **stage** (n = 165) | | | | | |
| I | 4 (36.4%) | 16 (25.4%) | 14 (15.4%) | 1.921[b] | 0.055 |
| II | 4 (36.4%) | 34 (54.0%) | 49 (53.8%) | | |
| III | 3 (27.2%) | 13 (20.6%) | 28 (30.8%) | | |
| **Histological grade** (n = 150) | | | | | |
| I ∼ II | 4 (40.0%) | 40 (67.8%) | 44 (54.3%) | −0.786[a] | 0.432 |
| III | 6 (60.0%) | 19 (32.2%) | 37 (45.6%) | | |
| **Vascular invasion** (n = 127) | | | | | |
| No | 8 (80.0%) | 29 (52.7%) | 23 (37.1%) | −2.559[a] | 0.011 |
| Yes | 2 (20.0%) | 26 (47.3%) | 39 (62.9%) | | |
| **Lymphatic vessel invasion** (n = 115) | | | | | |
| No | 10 (90.9%) | 32 (65.3%) | 22 (40.0%) | −3.523[a] | <0.001 |
| Yes | 1 (9.1%) | 17 (34.7%) | 33 (60.0%) | | |
| **nervous infiltration** (n = 113) | | | | | |
| No | 9 (81.8%) | 38 (76.0%) | 34 (65.4%) | −1.419[a] | 0.156 |
| Yes | 2 (18.2%) | 12 (24.0%) | 18 (34.6%) | | |

Notes.
[a] Mann–whitney $U$ test.
[b] Jonckheere-Terpstra test.

## Multivariate analysis

The factors, including tumor size, vascular invasion, lymphatic vessel invasion, and HER-2 expression status, were entered into the ordinal logistical analysis. Due to the missing data of clinicopathological features, 85 cases (50.3%) were excluded and 84 cases (49.7%) with complete data were finally included in the ordinal logistic regression analysis. The multivariate analysis result indicated that lymphatic vessel invasion was the only independent correlated factor of high fibrosis in FF (OR = 4.10, 95% CI 1.23 ∼13.70, $P = 0.021$).

**Table 3** The association analysis of different variables between molecular subtypes and degree of fibrosis in FF.

| Variables | Degree of fibrosis in FF | | | Z/J-value | P-value |
|---|---|---|---|---|---|
| | Mild n (%) | Moderate n (%) | High n (%) | | |
| **ER** (n = 168) | | | | | |
| Negative | 6 (54.6%) | 20 (31.2%) | 34 (36.6%) | −0.060[a] | 0.952 |
| Positive | 5 (45.4%) | 44 (68.8%) | 59 (63.4%) | | |
| **PR** (n = 168) | | | | | |
| Negative | 7 (63.6%) | 22 (34.4%) | 39 (41.9%) | −0.040[a] | 0.968 |
| Positive | 4 (36.4%) | 42 (65.6%) | 54 (58.1%) | | |
| **Ki-67** (n = 164) | | | | | |
| ≤14% | 3 (27.3%) | 16 (26.2%) | 16 (17.4%) | −1.377[a] | 0.168 |
| >14% | 8 (72.7%) | 45 (73.8%) | 76 (82.6%) | | |
| **HER-2** (n = 131) | | | | | |
| Negative | 9 (90.0%) | 39 (79.6%) | 47 (65.3%) | −2.141[a] | 0.032 |
| Positive | 1 (10.0%) | 10 (20.4%) | 25 (34.7%) | | |
| **Molecular subtypes** (n = 150) | | | | | |
| Luminal A | 1 (9.1%) | 9 (16.7%) | 6 (7.1%) | 0.572[b] | 0.568 |
| Luminal B | 6 (54.5%) | 33 (61.1%) | 54 (63.5%) | | |
| HER-2 overexpression | 0 (0.0%) | 2 (3.7%) | 15 (17.6%) | | |
| TNBC | 4 (36.4%) | 10 (18.5%) | 10 (11.8%) | | |

Notes.
[a] Mann–whitney $U$ test.
[b] Jonckheere-Terpstra test.

## Comparison of NPI variables between mild and moderate fibrosis in FF and high fibrosis in FF

A total of 161 patients with complete data were eligible for NPI scoring. The mean NPI was 4.60 ±1.40. Due to the small number of patients (6.5%) with mild fibrosis in FF, we combined the patients with mild fibrosis and those with moderate fibrosis into one group. Finally, we divided the 161 cases into two groups, mild and moderate fibrosis group and high fibrosis group. No significant difference was found between the two groups ($Z = −1.862$, $P = 0.063$) (Fig. 3). The cases were stratified by the clinicopathological characteristics to further analyze the differences of NPI variables between mild and moderate fibrosis group and high fibrosis group. In the no nerve infiltration subgroup, the no vascular infiltration subgroup, and the Luminal A subgroup, the NPI variables of high fibrosis in FF were significantly higher than those of mild and moderate fibrosis in FF ($P = 0.039$, $0.014$, and $0.018$; respectively) (Table 4 and Fig. 3).

## DISCUSSION

Our research found that tumor size >two cm, vascular invasion, lymphatic vessel invasion, and HER-2+ were positively correlated with the degree of fibrosis within FF in breast IDC by univariate analysis. Moreover, lymphatic vessel invasion was the only independent

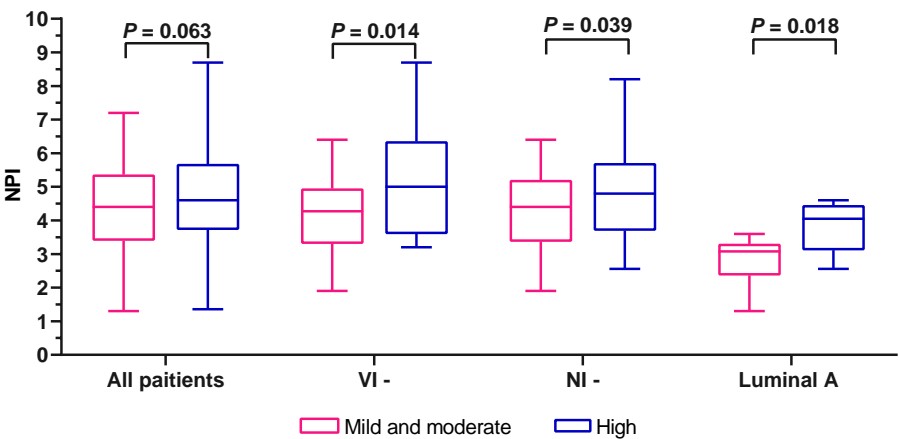

**Figure 3** **The comparison of NPI variables between mild and moderate fibrosis in FF and high fibrosis in FF.** Notes: VI-, no vascular invasion; NI-, no nervous infiltration.

correlation factor of high fibrosis in FF in breast IDC by multivariate analysis. Although there was no significant difference between the NPI of mild and moderate fibrosis in FF and that of high fibrosis in FF, further analysis showed that the NPI variables of high fibrosis in FF were significantly higher than those of mild and moderate fibrosis in FF in the no vascular infiltration subgroup, the no nerve infiltration subgroup, and the Luminal A subgroup. The above results indicated that the high fibrosis of FF was closely correlated with the aggressive clinicopathological characteristics of breast IDC and reflected the poor outcome of breast IDC with no vascular infiltration, no nerve infiltration or Luminal A subtype.

Hypoxia is the main cause of the fibrosis in tumor stroma (*Daniel et al., 2019*; *Hoffmann et al., 2018*). In hypoxia state, tumor cells secrete platelet derived growth factor (PDGF), transforming growth factor- $\beta$ (TGF- $\beta$), and fibroblast growth factor-2 (FGF-2). All these cell factors act on the fibroblasts together, resulting in the deposition and remodeling of extracellular matrix and the graduate formation of FF in the tumor stroma. Besides, PDGF, TGF- $\beta$, and FGF-2, together with vascular endothelia growth factor (VEGFR) secreted by fibroblasts and tumor associated macrophages, promote the angiogenesis, lymphangiogenesis, and lymphovascular invasion (*Shimada et al., 2017*; *Van den Eynden et al., 2007*). The carcinoma-associated fibroblasts (CAFs), which are the main tumor interstitial cells and transforms from fibroblasts, result in the deposition of collagen and fibronectin in the extracellular stroma. It is considered that CAFs play an important function in the formation of FF and are closely related to the high malignancy of tumor (*Balachander et al., 2018*; *Eiro et al., 2018*; *Reisfeld, 2013*; *Yang et al., 2016*). In addition, the fibrosis can inhibit T lymphocytes infiltration in tumor stroma and resist tumor immunity (*Salmon et al., 2012*). Therefore, FF forming in tumor stroma demonstrates the high aggressiveness of carcinoma (*Van den Eynden et al., 2008*). The tumor interstitial fibrosis associated with the adverse prognosis and anti-tumor drug resistance (*Grasso, Jansen &*

**Table 4  The comparison of NPI variables between mild and moderate fibrosis in FF and high fibrosis in FF after stratification.**

| Stratification | Degree of fibrosis in FF | | Z-Value[a] | P-value |
|---|---|---|---|---|
| | Mild and moderate $P_{25}$,M, $P_{75}$ (n) | High $P_{25}$, M, $P_{75}$ (n) | | |
| **Age (years)** | | | | |
| ≤50 | 3.36,4.50,5.40 (37) | 3.36,4.50,5.40 (37) | −0.930 | 0.352 |
| >50 | 3.40,4.40,5.20 (36) | 4.30,4.70,5.60 (51) | −1.690 | 0.091 |
| **Tumor size** | | | | |
| ≤2 cm | 3.30,4.27,5.20 (32) | 3.33,4.26,4.90 (25) | −0.299 | 0.765 |
| >2 cm | 3.60,4.60,5.60 (41) | 4.50,5.00,5.80 (63) | −1.448 | 0.148 |
| **Lymph node metastasis** | | | | |
| No | 3.29,3.50,4.40 (38) | 3.30,3.70,4.50 (35) | −0.696 | 0.486 |
| Yes | 4.60,5.36,5.90 (35) | 4.60,5.60,6.35 (53) | −1.063 | 0.288 |
| **stage** | | | | |
| I | 3.21,3.36,4.29 (20) | 3.30,3.40,4.27 (14) | −0.406 | 0.691 |
| II | 3.58,4.50,5.15 (37) | 3.70,4.54,5.00 (47) | −0.465 | 0.642 |
| III | 5.32,5.80,6.30 (16) | 5.60,6.30,6.80 (27) | −1.436 | 0.151 |
| **Histological grade** | | | | |
| I ~ II | 3.32,3.70,5.00 (43) | 3.50,4.50,5.40 (43) | −1.517 | 0.129 |
| III | 4.50,5.20,5.60 (25) | 4.57,5.60,6.45 (37) | −1.443 | 0.149 |
| **Vascular invasion** | | | | |
| No | 3.31,4.27,4.95 (36) | 3.60,5.00,6.35 (21) | −2.458 | 0.014 |
| Yes | 3.58,5.10,5.70 (26) | 4.31,4.93,5.85 (38) | −0.616 | 0.538 |
| **Lymphatic vessel invasion** | | | | |
| No | 3.36,4.40,5.18 (40) | 3.50,5.00,6.00 (21) | −1.709 | 0.087 |
| Yes | 3.58,4.56,5.70 (17) | 4.46,5.18,5.80 (32) | −1.441 | 0.150 |
| **Nervous infiltration** | | | | |
| No | 3.37,4.40,5.20 (45) | 3.70,4.80,5.70(32) | −2.063 | 0.039 |
| Yes | 3.35,4.80,5.65 (13) | 4.43,5.20,6.53 (16) | −0.430 | 0.446 |
| **ER** | | | | |
| Negative | 3.68,5.00,5.43 (26) | 4.53,5.40,6.30 (32) | −1.799 | 0.072 |
| Positive | 3.30,4.40,5.00 (47) | 3.50,4.47,5.60 (56) | −1.060 | 0.289 |
| **PR** | | | | |
| Negative | 4.28,4.68,5.39 (28) | 4.57,5.10,6.20 (37) | −1.955 | 0.051 |
| Positive | 3.30,3.90,5.25 (45) | 3.40,4.40,5.60 (51) | −0.962 | 0.336 |
| **Ki-67** | | | | |
| ≤14% | 2.77,3.40,4.42 (17) | 3.33,3.70,4.58 (16) | −1.353 | 0.179 |
| >14% | 3.65,4.60,5.40 (53) | 4.26,4.96,5.70 (71) | −1.519 | 0.129 |
| **HER-2** | | | | |
| Negative | 3.39,4.40,5.32 (46) | 3.70,4.50,5.70 (43) | −1.450 | 0.147 |
| Positive | 4.56,5.20,5.70 (11) | 4.32,5.05,5.80 (24) | −0.124 | 0.903 |

**Table 4** (*continued*)

| Stratification | Degree of fibrosis in FF | | Z-Value[a] | P-value |
|---|---|---|---|---|
| | Mild and moderate $P_{25}$,M, $P_{75}$ (*n*) | High $P_{25}$, M, $P_{75}$ (*n*) | | |
| **Molecular subtypes** | | | | |
| Luminal A | 2.37,3.08,3.30 (9) | 3.12,4.05,4.45 (6) | −2.359 | 0.018 |
| Luminal B | 3.48,4.58,5.43 (38) | 3.70,4.50,5.60 (51) | −0.183 | 0.855 |
| HER-2 overexpression | 5.15[b] (2) | 4.93,5.70,6.58 (14) | −0.954 | 0.417 |
| TNBC | 3.68,4.50,5.40 (14) | 4.10,5.00,7.55 (9) | −1.073 | 0.305 |

**Notes.**
[a] Mann–whitney $U$ test.
[b] 5.15 was the median NPI.
Since there were only 2 cases, the $P_{25}$ and $P_{75}$ could not be calculated.

*Giovannetti, 2017*) has been observed in the carcinoma of breast cancer, pancreatic cancer (*Thomas & Radhakrishnan, 2019*), colorectal cancer (*Ikuta et al., 2018*), and gastric cancer.

*Hasebe et al. (1996)* demonstrated that the histological grade and lymph node metastasis rate were higher in breast cancer with FF, especially in that with tumor size<five cm. *Jeong et al. (2018)* revealed that the FF was significantly associated with tumor size >two cm, lymph node metastasis, poor differentiation, and vascular invasion in breast cancer. The studies mentioned above had indicated that FF was closely related with the adverse pathological characteristics of breast cancer. However, previous studies had rarely reported the association between the degree of fibrosis in FF and the pathological features of breast cancer, so we conducted this research. Our study found that the degree of fibrosis within FF in breast IDC with tumor size >2 cm was significantly higher than that with tumor size ≤ two cm, indicating the more severe hypoxia occurred in larger tumors. Besides, our study also revealed that the degree of fibrosis within FF was significantly higher in breast cancer with vascular infiltration and lymphatic infiltration. It might be that the densities of blood vessels and lymphatics were higher in the tumor with high degree fibrosis of FF, increasing the probability of lymphovascular invasion. The result of multivariate analysis showed that lymphangitic infiltration was the only independent factor correlating with the high fibrosis in FF in patients with IDC. Due to the missing data of clinicopathological features, only 49.7% cases with complete data were finally included in the ordinal logistic regression analysis, which might prevent us from discovering more factors correlating with the high fibrosis within FF in multivariate analysis. To sum up, the high fibrosis of FF can predict the strong invasiveness and high malignancy of breast IDC.

The relation between FF and the sub-factors of molecular subtypes is not very clear now. *Hasebe et al. (2000)*; *Hasebe et al. (1996)* showed that FF was significantly correlated with HER-2 protein overexpression in IDC. *Mujtaba et al. (2013)* revealed that FF was negatively associated with HER-2 expression ($P = 0.021$) and Ki-67 index ($P = 0.001$) and positively associated with HR expression ($P = 0.007$). However, *Jeong et al. (2018)* did not find that FF was related to the expression of HR, HER-2, and Ki-67. The results of the studies above were controversial, which may be caused by the followings. Firstly, there were other histopathological types of breast cancer besides IDC, which were included in Mujtaba's

and Jeong's studies. And the molecular subtypes of varying histopathological types were distinct in breast cancer. Moreover, the testing methods and the diagnostic criterion of HER-2 expression and Ki-67 index were different from the current ones. The association between FF and molecular subtypes is also unclear. *Mujtaba et al. (2013)* demonstrated that the FF was more common in Luminal A subtype than in non-Luminal A subtype ($n = 450$, $P < 0.001$). But another study (*Jeong et al., 2018*) did not find the association between FF and molecular subtypes ($n = 291$, $P = 0.830$). The different conclusions might be due to the inconsistencies of baseline characteristics of the cases included in the two studies. So more studies are needed in exploring the association between FF and molecular subtypes of breast cancer.

Few studies were conducted on the relation between the degree of fibrosis in FF and the sub-factors of molecular subtypes of breast cancer. Only H (*Hasebe et al., 1996*) revealed that the HER-2 protein significantly overexpressed in the breast cancer with moderate and high fibrosis in FF (90.9% vs. 41.7%, $n = 153$, $P < 0.02$), compared with the breast cancer with mild fibrosis in FF. The similar conclusion was drawn in our study (97.2% vs. 90.5%, $n = 131$, $P = 0.032$). Thus it can be seen that moderate and high fibrosis of FF is correlated with the invasiveness of breast cancer. Our study did not reveal any associations between the degree of fibrosis and other sub-factors of molecular subtypes including ER, PR and Ki-67 index. In addition, our result revealed that no association was found between the degree of fibrosis in FF and molecular subtypes. The relation between the degree of fibrosis in FF and molecular subtypes was not investigated in previous studies, so more studies are needed to explore the question.

The associations between FF and the survival of breast cancer have been reported in many studies. *Hasebe et al. (1998)* revealed that the presence of FF predicted higher risk of recurrence and death in breast cancer with less than four lymph nodes metastases or stage I $\sim$ II B, compared with the absence of FF. Another study (*Colpaert et al., 2001*) including 104 cases of breast cancer with stage $T_{1\sim2}N_0M_0$ showed that the median disease-free survival (DFS) of cases with the presence of FF was significantly shorter than that with the absence of FF (25.0 months versus 91.5 months, $P < 0.05$). *Shimada et al. (2017)* demonstrated that the higher risk of recurrence (HR, 7.8; 95%CI [2.6–22.8]; $P < 0.001$) and the shorter median progression-free survival (PFS) were found in breast cancer with the presence of FF. The studies above had indicated that the presence of FF was associated with the poor outcome of breast cancer. Therefore, FF was considered as a significant prognostic feature for breast cancer (*Hasebe et al., 1998*; *Mujtaba et al., 2013*). On the basis of previous studies, we further explored the correlation between the degree of fibrosis in FF and the long-term survival of breast IDC. All the cases included in our study had not been followed up to the median survival time. Thus, the short follow-up period led to the current inability of survival analysis, which was the greatest deficiency of our study. NPI was adopted as an alternative survival indicator to investigate the relation between the degree of fibrosis in FF and the long-term survival of breast cancer. NPI, reported first by *Lee & Ellis (2008)*, is used widely to predict the 10-year overall survival (OS) rate of early breast cancer. NPI is a scoring system of clinicopathology containing tumor size, histological grade, and lymph node stage. The NPI scores of 2.02 $\sim$2.40, 2.41 $\sim$3.40, 3.41 $\sim$4.40, 4.41 $\sim$5.40, 5.41 $\sim$6.40,

and 6.41 ∼6.80 predict the 10-year overall survival rates of 96%, 93%, 81%, 74%, 55%, and 38%, respectively. According to the cutoffs of NPI score 3.40 and 5.40, the early breast cancer patients are stratified into good, moderate, and poor groups. The mean NPI in this study was 4.60 ±1.40 and most cases were evaluated as moderate prognosis. Our study revealed that the NPI of high fibrosis in FF showed an upward tendency, compared with that of mild and moderate fibrosis in FF ($P = 0.063$). Further stratified analysis found that the NPI of high fibrosis in FF was significantly higher than that of mild and moderate fibrosis in FF in the no vascular infiltration subgroup, the no nerve infiltration subgroup, and the Luminal A subgroup. In general, the clinicopathological features of no vascular infiltration, no nerve infiltration, and Luminal A subtype indicate favorable outcomes of breast IDC. The presence of FF with high fibrosis could indicate the relatively worse outcomes of cases with the favourable clinicopathological features of breast IDC mentioned above, which was found in our study. Thus, it can be seen that the degree of fibrosis in FF could be used as a practical and meaningful pathological feature for predicting the survival outcome of early breast IDC.

## CONCLUSIONS

In summary, our study demonstrated that the high fibrosis in FF was closely associated with the strong invasiveness and the high malignancy of breast IDC. There are some disadvantages in our study, including the small sample size, partial clinicopathological data missing, and short follow-up period. In the future study, expanding the sample size, collecting sufficient clinicopathological data, and extending follow-up time should be considered.

## ACKNOWLEDGEMENTS

We thank Doctor Rongxin Yan for assistance of collecting pathological data, and thank Doctor Xiangtao Lin for taking the pathological pictures.

### Funding

This study was funded by the National Natural Science Foundation of China (Grant NO. 81301912), the Beijing Municipal Health System High-level Health Person Foundation Project (Grant NO. 2014-3-005), and the Beijing Municipal Science and Technology Commission Foundation (Capital Features, Z161100000516083). The funders had no role in study design, data collection and analysis, decision to publish, or preparation of the manuscript.

### Grant Disclosures

The following grant information was disclosed by the authors:
National Natural Science Foundation of China: 81301912.

Beijing Municipal Health System High-level Health Person Foundation Project: 2014-3-005.

Beijing Municipal Science and Technology Commission Foundation (Capital Features, Z161100000516083).

## Competing Interests

The authors declare there are no competing interests.

## Author Contributions

- Yongfu Li conceived and designed the experiments, performed the experiments, analyzed the data, contributed reagents/materials/analysis tools, authored or reviewed drafts of the paper, approved the final draft.
- Yuhan Wei analyzed the data, prepared figures and/or tables, approved the final draft.
- Wenjun Tang conceived and designed the experiments, prepared figures and/or tables, approved the final draft.
- Jingru Luo performed the experiments, prepared figures and/or tables, approved the final draft.
- Minghua Wang performed the experiments, contributed reagents/materials/analysis tools, authored or reviewed drafts of the paper, approved the final draft.
- Haifeng Lin analyzed the data, prepared figures and/or tables, approved the final draft.
- Hong Guo and Yuling Ma analyzed the data, contributed reagents/materials/analysis tools, prepared figures and/or tables, approved the final draft.
- Jun Zhang conceived and designed the experiments, authored or reviewed drafts of the paper, approved the final draft.
- Qin Li conceived and designed the experiments, contributed reagents/materials/analysis tools, authored or reviewed drafts of the paper, approved the final draft.

## Human Ethics

The following information was supplied relating to ethical approvals (i.e., approving body and any reference numbers):

The Medical Ethics Committee of the Second Affiliated Hospital of Hainan Medical University granted ethical approval to carry out the study within its facilities (Ethical Application Ref:LW005).

## Data Availability

The raw data are available in the Supplemental File.

## Supplemental Information

Supplemental information for this article can be found online at http://dx.doi.org/10.7717/peerj.8067#supplemental-information.

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
