# Peer review of "Association between the degree of fibrosis in fibrotic focus and the unfavorable clinicopathological prognostic features of breast cancer"

_PeerJ, doi:10.7717/peerj.8067_

## Round 0.1 · original submission · Minor Revisions

Dear Mr. Li,

I think it would be extremely important to answer the questions and observations of the reviewers. Particularly note the question under number three (Rev 2).

Hopefully there will be enough time for a quality response.

Prof Nebojsa Arsenijevic

Reviewer 1 ·

Basic reporting

No comment

Experimental design

No comment

Validity of the findings

No comment

Additional comments

In the Manuscript entitled “Association between the degree of fibrosis in fibrotic focus and the prognosis of breast cancer” the authors provided evidences that high fibrosis in tumor microenvironment fibrotic focuses correlated with the aggressiveness of breast invasive ductal carcinoma, further indicating that the degree of fibrosis in fibrotic focuses might be a useful marker of the prognosis of breast cancer. However, as authors noted in the text, some limitations of the study should be considered in future research in this field.
Taken together, the paper is interesting and well-written thus providing some novel insights into the topic. The Manuscript is recommended for publication.

Reviewer 2 ·

Basic reporting

No comment
There is some information missing in the raw data, which is not of importance for the final results (family tumor histories, drinking and smoking history).

Experimental design

No comment.

Validity of the findings

The impact and novelty of this manuscript cannot be assessed. However, this research has a very interesting theme that is very scarce in modern literature. The researched connection between clinico-pathological prognostic features in breast cancer is presented systematically in a very interesting way.The acquired results are potentially of great interest.

Additional comments

1. The authors state in the manuscript (line 82-84): “Therefore, a retrospective study was conducted to explore the association between the degree of fibrosis in FF and the prognosis of breast cancer.” The follow up patients was too short to speak about breast cancer prognosis. Maybe it would be more adequate to speak about the connection between the FF and unfavorable clinicоpathological prognostic features.
2.The authors cite a very interesting statement in their manuscript (line 146-147): “In addition, the densities of micro-vessels and micro-lymphatic vessels in FF were significantly higher than those in normal tissue.” Could the research also include the analysis of microvascular density in FF and peritumor tissue? If so, what are the reasons for not including this characteristic in the research?
3. I ask the authors to explain why the FF size was not measured in the research? Hypothetically, the correlation between the degree of fibrosis in FF and the size of FF could maybe give an answer to the crucial question wich of these two parameters has a higher significance.
4.In the raw data 38 BC with equivocal HER-2 status exist.Please explain. Why was the HER-2 status in these patients remained equivocal(2+)? Would the final defining of HER-2 status in these BC change the statistic results ad connection between the fibrosis in FF and HER-2 positivity in BC?
5. In the raw data, in sections P and Q exist 9 BC who have ER-PR+ status. They are mainly classified in the Luminal B breast cancer group. Was 20% of positive tumor cells a limit for PR positivity in the both Luminal BC groups (Luminal A and Luminal B)?

Reviewer 3 ·

Basic reporting

This was a generally well conducted study implicating close association between high fibrosis in fibrotic focus in breast invasive ductal carcinoma and invasiveness and the high malignancy of breast.
Manuscript is clearly written, the aim of the research clearly emerges from the background section. The references are relevant and well stated.

Experimental design

This is original research within aims and scope of the journal. Research question is well defined, investigation is well performed, methods are described with enough details. There is no data about degree of fibrosis in fibrotic focus and prognosis of breast cancer.

Validity of the findings

Research question is well defined, investigation is well performed, methods are described with enough details. Conclusion fits the obtained data.

Additional comments

Minor points:
Arrows that indicate main findings listed in figure legend should be inserted in Figure 1.
There are missing or extra spaces in lines 66, 71, 72, 148, 247, 248, 272.
It is stated that fibroblastic proliferation can be seen in Fig 1C (stated in figure legend), I suggest high number of fibroblasts instead fibroblastic proliferation, since proliferation of cells can not be detected by H&E staining.

---

## Round 0.2 · accepted · Accept

Dear Mr. Li,

I carefully looked at all the changes you made. You have answered all the questions and made all the necessary changes and it is now acceptable to publish in PeerJ. Congratulations.

Nebojsa Arsenijevic